# Acute Kidney Injury in Kidney Transplant Patients in Intensive Care Unit: From Pathogenesis to Clinical Management

**DOI:** 10.3390/biomedicines11051474

**Published:** 2023-05-18

**Authors:** Marco Fiorentino, Francesca Bagagli, Annamaria Deleonardis, Alessandra Stasi, Rossana Franzin, Francesca Conserva, Barbara Infante, Giovanni Stallone, Paola Pontrelli, Loreto Gesualdo

**Affiliations:** 1Nephrology, Dialysis and Transplantation Unit, Department of Precision and Regenerative Medicine and Ionian Area (DiMePRe-J), University of Bari “Aldo Moro”, 70121 Bari, Italy; marco.fiorentino@uniba.it (M.F.);; 2Nephrology, Dialysis and Transplantation Unit, Department of Medical and Surgical Science, University of Foggia, 71122 Foggia, Italy

**Keywords:** acute kidney injury, kidney transplantation, intensive care unit, sepsis, immunosuppression

## Abstract

Kidney transplantation is the first-choice treatment for end-stage renal disease (ESRD). Kidney transplant recipients (KTRs) are at higher risk of experiencing a life-threatening event requiring intensive care unit (ICU) admission, mainly in the late post-transplant period (more than 6 months after transplantation). Urosepsis and bloodstream infections account for almost half of ICU admissions in this population; in addition, potential side effects related to immunosuppressive treatment should be accounted for cytotoxic and ischemic changes induced by calcineurin inhibitor (CNI), sirolimus/CNI-induced thrombotic microangiopathy and posterior reversible encephalopathy syndrome. Throughout the ICU stay, Acute Kidney Injury (AKI) incidence is common and ranges from 10% to 80%, and up to 40% will require renal replacement therapy. In-hospital mortality can reach 30% and correlates with acute illness severity and admission diagnosis. Graft survival is subordinated to baseline estimated glomerular filtration rate (eGFR), clinical presentation, disease severity and potential drug nephrotoxicity. The present review aims to define the impact of AKI events on short- and long-term outcomes in KTRs, focusing on the epidemiologic data regarding AKI incidence in this subpopulation; the pathophysiological mechanisms underlying AKI development and potential AKI biomarkers in kidney transplantation, graft and patients’ outcomes; the current diagnostic work up and management of AKI; and the modulation of immunosuppression in ICU-admitted KTRs.

## 1. Introduction

Kidney transplantation (KT) is the best therapeutic option for patients with end-stage renal disease (ESRD) and is associated with a better quality of life and lower mortality compared to patients on chronic dialysis [1]. Advances in the immunosuppressive management of such patients have led to improvement in the overall graft survival and have increased the number of transplantations performed at high immunological risk [2]; however, these patients frequently experienced short- and long-term complications, and about 10% of kidney transplant recipients (KTRs) presented life-threatening conditions requiring intensive care unit (ICU) admission, mainly related to acute ventilatory failure and sepsis/septic shock [3]. In this scenario, KTRs are at higher risk of AKI onset during ICU stay, and a discrete proportion will require renal replacement therapy (RRT). AKI is now recognized as a leading cause of short-term outcomes among critically ill patients, as AKI onset and progression is associated with high risk for mortality and chronic comorbidities, such as chronic kidney disease (CKD) and cardiovascular diseases [4,5,6,7,8]. KTRs that develop AKI have a relative risk of graft loss; renal outcomes in such populations result from the combination of the pre-existing CKD (i.e., baseline renal function), the use of nephrotoxic agents and episodes of hemodynamic instability during ICU stay [9,10,11]. Moreover, the need to reduce or withdraw the immunosuppressive treatments may induce additional immunological injuries, leading to the progression of graft dysfunction due to the development of de novo donor-specific antibodies (DSA) during or after ICU stay, with a consequent reduced graft survival among ICU survivors [12,13]. To date, epidemiological data on AKI in KTRs are limited, and no specific recommendations for the management of KTRs with AKI are available. Here, we aim to summarize the main epidemiological data of ICU admission and AKI incidence, the pathophysiology of AKI in this specific population given their different immunological statuses, the potential role of AKI biomarkers in this setting, as well as the management of immunosuppression during ICU stay.

## 2. Epidemiology of ICU Admission and AKI Occurrence in KTRs

AKI is estimated to occur in one fifth of the adult general population with acute illness, further increasing during ICU stay [14]. Kidney transplantation is considered to be a risk factor for AKI. Predisposing factors include variable degrees of ischemia-reperfusion injury upon organ withdrawal and transplantation, chronic use of nephrotoxic calcineurin inhibitors and the condition of single-functioning organ [15,16]. Data about lifetime incidence of AKI in KTRs are poor and inconclusive. ICU admission is estimated to be around 6% of KTRs in the delayed post-transplant period [3], although higher ICU admission rates were reported in the literature and secondary to hospital-specific post-transplant protocols, in which ICU admission represents the step-down following transplantation. ICU admission is considered an independent risk factor for AKI development and graft survival in KTRs, particularly in the late post-transplant period (>6 months after kidney transplantation), wherein calcineurin inhibitor (CNI) nephrotoxicity plays a major role in pathogenesis [3,11,17]. AKI is reported to occur more frequently in ICU than in non-ICU hospitalizations, and fatality rates might reach up to 30% [3,18]. As shown in Table 1, AKI incidence broadly varies in this specific population, ranging from 10 up to almost 80% [3,18,19]. Mehrotra et al. systematically analyzed the incidence and outcomes of AKI amongst more than twenty-seven thousand Medicare-insured KTRs between 6 months and 3 years after transplant, reporting an incidence of AKI in 11.3% of cases, regardless of RRT requirement [19]. However, significant differences in the post-transplant ICU admission protocols, and socio-economic disparities between high-income and low- and middle-income countries are potential biases in interpreting such data [3,20]. Concerning the socio-economic settings, low- and middle-income countries are burdened by higher rates of community-acquired AKI, advanced AKI stages and worse kidney outcomes with respect to high-income countries, even though data are suboptimal and standardized AKI staging assessments are lacking [21].

Furthermore, the recent COVID-19 pandemic contributed to a boost in AKI incidence [22]: it is estimated that AKI occurred in more than 80% of COVID-19-infected KTRs. Aziz et al. reported that KTRs with COVID-19 infection have a 2-fold increased risk of AKI, prolonged ICU stay and death, and a 4-fold increase for RRT [23]. The severity of infection was correlated with a worse graft outcome and increased fatality rate (26% [95% CI 0.22–0.32]) [24,25]. Worth noting, preliminary reports showed differences in kidney outcomes and mortality between the first and the second wave of pandemic. In 2020, the TANGO International Consortium described an increased risk of AKI and mortality among 9845 hospitalized KTRs with COVID-19 infection, reaching 52% and 32%, respectively [26]. However, a French multicentric meta-analysis argues against these findings, describing comparable outcomes for both KTRs and nontransplant patients when matched for comorbidities [27]. Although these data may potentially raise questions regarding an improvement in the management of transplanted patients amidst the two waves, the fatality rates were similar between different pandemic waves (26% vs. 23%) in a Dutch meta-analysis including 179 studies including COVID-19 positive KTRs [25].

**Table 1 biomedicines-11-01474-t001:** Main studies focusing on AKI incidence and outcomes in kidney transplant recipients.

Study ID	Year	Study Design	Patients	AKI Incidence	Mortality Rate	Additional Epidemiological Findings
Mehrotra et al. [19]	2012	Retrospective longitudinal cohort study; AKI incidence and outcomes in KTRs	27,232	11.6%	5.8%	AKI is an independent factor for graft loss (HR 2.74), death with functioning graft (HR 2.36) and graft loss (HR 3.17). AKI paradoxically associates with worse outcomes in early CKD stages.
Filiponi et al. [11]	2015	Single-center, retrospective cohort study; 1-year graft survival in KTRs with AKI	458	82.3%	2.1%	CMV infection being the most common cause of hospitalization (20.3%), followed by urosepsis (14.4%). ICU admission OR: 8.9; contrast media use OR: 9.34.
Panek et al. [17]	2015	Single-center retrospective cohort study; clinical outcomes of KTRs at 1 year post-transplantation	326	21.0%	1.2 deaths/100 PY	CNI toxicity is the leading cause of AKI (33%). The presence of AKI does not have any impact on mortality rate.
Guinault et al. [18]	2019	Multicenter, retrospective observational study; outcomes in ICU admitted KTRs	200	85.1%	26.5%	Death occurring mostly within the first 6 months. CKD progression observed in 45.1% of survivors; 15.1% developed new anti-HLA antibodies.
Cravedi et al. [26]	2020	International, multicenter, retrospective cohort study; clinical outcomes in COVID-19-positive KTRs	144	51.0%	32.0%	AKI occurred in 52% cases, with respiratory failure requiring intubation in 29%, and the mortality rate was 32%. Risk factors for mortality: older age, lower lymphocyte counts and baseline eGFR, higher serum lactate dehydrogenase, procalcitonin and IL-6.
Camargo-Salamanca et al. [16]	2020	Retrospective cohort study; AKI incidence and risk factors	179	58.1%	3.9%	KTRs with higher baseline serum creatinine (OR, 2.6; 95% CI 1.5–4.5, *p* < 0.001) and hospital admission because of infections (OR, 2.4; 95% CI, 1.1–5.2; *p* = 0.020) were more likely to experience AKI. 19 recipients (10.6%) had graft loss with a significant AKI association (*p* = 0.003).
Kremer et al. [25]	2021	Meta-analysis; clinical outcomes in COVID-19-infected KTRs	5559	50.0%	23.0%	Mortality rates are significantly increased in the early post-transplantation period (15 months post-TX). No differences are reported in AKI risk between early and late post-transplantation periods.

**Legend:** AKI: acute kidney injury; ARF: acute respiratory failure; CKD: chronic kidney disease; CMV: cytomegalovirus; HLA: human leukocyte antigen; HR hazard ratio; ICU: intensive care unit; KTR; kidney transplant recipient; OR: odds ratio; PY: patient-years; RRT: renal replacement therapy; sCr: serum creatinine; TX: transplantation.

## 3. Causes of AKI among KTRs

AKI consists of an abrupt worsening of the renal function, characterized by an increase in serum creatinine by at least 0.3 mg/dL within 48 h, an increase in serum creatinine to 1.5 times from baseline within a week or urinary output inferior to 0.5 mL/kg/h for 6 h, according to KDIGO criteria [28]. AKI is a multifactorial disease and has its roots in chronic renal insults, primarily due to cardiovascular and chronic kidney disease (CKD) by any cause. It is widely acknowledged that CKD is an independent risk factor for AKI development, in which a trigger may precipitate a previously latent renal insufficiency. However, amongst acute triggering events, clinical conditions characterized by hypovolemia, infections/sepsis events and drug toxicity are recognized as the most common causes of AKI in KTRs (Figure 1). Early recognition and diagnosis is often complicated, as the differential diagnosis is quite wide, including immunological complications as well as recurrence of primary renal disease, and thus requires specific examination (graft biopsy) [12].

### 3.1. Infections

Infections and severe sepsis are universally considered the prominent cause of ICU admission for AKI in KTRs. Their incidence varies according to the timing after transplantation, with a prevalence of opportunistic infections and activation of latent infection during the first 12 months, while community-acquired infections predominate in the late post-transplant period (Figure 1) [29]. Infections after kidney transplantation may be challenging to diagnose and manage since these episodes may be clinically different in this specific immunosuppressed population [30]. Here, we reported the main clinical infectious disease reported in KTRs.

#### 3.1.1. Urinary Tract Infections

Urosepsis and urinary tract infections (UTIs) are the most frequent infectious complications and the main causes of AKI in this population, mostly occurring within the first six months post-transplantation, presumably due to the intense immunosuppressive treatment regimen, the presence of ureteral stents and the unique anatomical configuration of the graft, including a shorter ureter, lacking anti-reflux properties and denervation that may hinder an early diagnosis [31,32]. Urosepsis and UTIs occurrence impinge upon graft survival and mortality, although there is insufficient data to determine the real burden. An increased risk of 1-year graft loss and/or worsening of graft function in KTRs with AKI secondary to UTIs, compared to transplanted patients who did not develop AKI, has been reported [31]. Data regarding the prevalence of etiologic agents are limited; however, bacterial infections are predominant, particularly related to Gram-negative bacteria. *Escherichia coli* and Klebsiella pneumoniae are described as the principal infectious agents in KTRs with UTIs, although an increased incidence of infections due to multi-resistant organisms (Enterobacteriaceae, Pseudomonas species) and fungi (Candida species) have reported in recent years [33,34,35]. Acute graft pyelonephritis is a complication of UTIs, and its risk is closely correlated to the frequency of UTI recurrence [3]. In the ICU setting, acute graft pyelonephritis is the leading cause of around 20% of sepsis cases, and AKI occurrence is typically linked to the development of septic shock [9]. It is an independent risk factor for estimated glomerular filtration rate (eGFR) decline (−2.29 mL/min/1.73 m^2^ [95% CI: −3.23 to −1.35]) and graft loss at 1 year (HR: 1.66 [95% CI: 1.05–2.64], although late (i.e., more than 6 months following transplantation) recurrences did not exert a cumulative detrimental effect on allograft survival or on acute rejection frequency [36,37,38]. Although several risk factors for UTIs are widely reported, including donor-related, recipient-related (female gender, advanced age, pre-transplant anatomic abnormalities, diabetes) and transplant-related factors (overimmunosuppression, prolonged placement of ureteral stent), a growing body of evidence highlighted the relevance of the urinary microbiome in the setting of UTIs recurrence and AKI episodes in KTRs [34]. In a recent cross-sectional pilot study, urinary microbiome dysbiosis was pointed out as a potential major player in the pathophysiology of urosepsis and UTIs in KTRs during AKI [39]. Human urobiome geno-phenotypic differences assessed by 16S microbial ribosomal gene sequencing were demonstrated to negatively impact graft function and survival as compared to non-transplanted patients, thus raising questions regarding the potential predictive value of the urinary microbiome and metabolome in AKI pathogenesis [39]. Microbiome changes in specific populations such as Firmicutes and Proteobacteria mostly occur within six months following transplantation, and they are supposed to favor AKI onset and impact kidney transplantation outcomes, altering the host’s immune system, inflammatory cytokines and production of uremic toxins [40,41,42].

#### 3.1.2. Bloodstream Infections

The second infectious cause of AKI in KTRs admitted in ICU is bloodstream infections with a cumulative incidence estimated between 10% and up to 40% of cases [9,43]. Discrepancies in the literature mostly arise from significant differences among diagnostic criteria and inpatient management protocols [44]. Apart from comorbidities and graft characteristics (diabetes mellitus, baseline eGFR, donor age, delayed graft function), UTI susceptibility, chronic immunosuppression and gut disturbances are considered as the main risk factors for bloodstream infections within this population; the development of shock and the need of mechanical ventilation are independent risk factors for mortality [45]. Abdomen and the urinary tract represent the primary source of infection in two-thirds of cases; bacterial infections are the leading cause of bloodstream infection in KTRs, with Gram-negative bacilli (*Escherichia coli*) as the prevalent community-acquired pathogen [46,47].

Cytomegalovirus (CMV) is ascribed as the leading viral pathogen in KTRs, particularly in the first months after kidney transplantation. CMV infection incidence is estimated to range between 40% and 80% [48]. CMV has multi-organ tropism, and clinical manifestation ranges from asymptomatic disease to severe forms with gastrointestinal, pulmonary and kidney involvement, representing an important risk factor for graft dysfunction. Mycophenolate mofetil use, lymphocytopenia, anti-thymocyte globulin immunosuppressive regimen, anti-CMV IgG seropositivity of the donor in a seronegative recipient and advanced age are the main risk factors associated with CMV reactivation after kidney transplantation [49].

Finally, chronic immunosuppressive therapy may lead to intestinal disturbances and gut microbiome modifications with consequent bacterial translocation. This iatrogenic dysbiosis may favor the proliferation of pathobionts (i.e., *Escherichia coli*) with subsequent release in the bloodstream of gut-derived uremic toxins (p-cresyl sulfate and indoxyl sulfate), aberrant short chain fatty acids and lipopolysaccharides, thus paving the way for inflammation-driven AKI and progression to CKD [50].

#### 3.1.3. Respiratory Tract

Lower respiratory tract infections are an uncommon complication of KTRs, typically in the setting of long-standing immunosuppressive therapy regimens, and represent the primary source of infection in KTRs in the ICU, accounting for around 50–60% of sepsis cases [10]. ICU management of pneumonia required intubation and mechanical ventilation in up to 60% of patient with a significant mortality risk (30–35%) [3,10]. In such conditions, AKI occurrence is reported in almost 15–25% of cases, with a consequent increased risk of mortality during hospitalization at 3 months and at 1 year in both severe and non-severe septic pneumonia [51]. In a French multicentric cohort study, a core respiratory pathogen has been identified in 61% of cases, among which Haemophilus influenzae prevailed (12%), while 38.7% included opportunistic infections headed by Pneumocystis jirovecii (12% of cases) [52]. The latter typically occurs in the late post-transplant period (after trimethoprim-sulfamethoxazole prophylaxis discontinuation) and after increased immunosuppression due to acute rejection episodes [10]. Finally, COVID-19 infection has evolved as a new causative agent for AKI in KTRs after the development of acute respiratory distress syndrome (ARDS) and ICU admission. Acute tubular injury, renal tissue hypoxia, thrombotic microangiopathy and direct cytotoxicity of kidney epithelial cells have been identified as predominant COVID-19-related histological findings [53,54,55]. COVID-19 KTRs reported a 3.55-fold increase in mortality with respect to KTRs not infected with COVID-19. Likewise, AKI incidence was increased in the transplanted population (50–75%), associated with a significant increase in mortality compared to the non-transplanted, age and comorbidity-matched counterpart and KTRs who did not experience AKI [25,56,57].

### 3.2. Drug Nephrotoxicity

The use of nephrotoxic medications is a well-known risk factor for AKI in KTRs. However, data in the literature regarding the global burden of nephrotoxic drugs on KTRs are sparse and mostly focus on single agents only. Calcineurin inhibitors (CNIs) are the milestone of immunosuppressive therapy in kidney transplantation, albeit burdened by a certain side effect. Dysregulation of vasoconstrictor factors, such as endothelin and thromboxane A2, associated with the reduced release of vasoactive molecules, including nitric oxide, prostacyclin and prostaglandin E2 in turn associated with both the renin–angiotensin–aldosterone system and sympathetic nervous system upregulation, led to abnormal vasoconstriction of the afferent arteriole and progressive allograft dysfunction [58,59]. Acute CNI toxicity is characterized by the presence of acute tubular necrosis, arteriolar hyalinosis of the tunica media, cell vacuolization and shedding in the proximal tubule and interstitial edema, with a reversible, dose-dependent decrease in eGFR [59,60]. Thrombotic microangiopathy (TMA) is a rare complication of acute sirolimus/CNI-related toxicity. Post-transplant TMA prevalence is estimated to range from 0.8% to 14%, and the initial use of sirolimus (SRL) is considered to be a risk factor; other potential causes of TMA in this setting are recurrence of atypical hemolytic-uremic syndrome, CMV infection or episodes of acute rejection. The peak of incidence mainly occurs within the first three months post-transplantation [61]. It is a life-threatening complication, which may be systemic or kidney-limited, and it is characterized by endothelial dysfunction, vasoconstriction and diffuse small-vessel thrombosis secondary to prothrombotic derangements [62]. The SRL/CNI-related form is histologically indistinguishable from other TMA etiologies [60,61,62,63]. Drug-induced TMA negatively affects graft and patient survival, wherein graft loss occurs in 33–40% of systemic cases at 2 years, and KTR survival rate at 3 years is estimated to be 50% [61,62,63,64]. SRL use is also associated with pulmonary toxicity, characterized by bilateral infiltrates and ground-glass opacities at chest radiography [65]. The incidence of SRL-associated pneumonitis is around 10% of KTRs in treatment with SRL. Although the severity of such a condition is usually mild, cases characterized by extensive lung infiltration associated with ARDS have been reported, requiring ICU admission and with increased risk of AKI episodes. Furthermore, drug-induced neurological complications are rarely reported as causes of ICU admission in KTRs. A rare radiological picture of posterior reversible encephalopathy syndrome (PRES), characterized by seizures, impaired consciousness and hypertension has been described in 0.3% of KTRs; CNI and SRL may represent risk factors for PRES, even in the presence of blood levels in the therapeutic range [66].

Finally, antimicrobials are routinely implemented, mostly in combinations, in the ICU setting, therefore acting as potent triggers for drug-induced AKI. The nephrotoxicity of aminoglycosides has been widely reported in literature, and they are acknowledged to upregulate reactive oxygen species (ROS) production on epithelial cells at the level of the S1 and S2 segments of the proximal tubule in a dose-dependent manner [67]. Toxicity manifests through progressive eGFR decline, hypo-osmotic polyuria and proteinuria [68,69]. Vancomycin-associated nephrotoxicity remains uncertain, with an incidence rate varying up to 40% [70]. Experimental models have shown the nephrotoxic effects of vancomycin, inducing epithelial cell proliferation at the proximal tubule level in a dose-dependent manner and the upregulation of ROS production; several studies reported acute tubular necrosis (ATN) on renal biopsy specimens, although renal biopsy is not routinely performed [70]. Amphotericin B-induced nephrotoxicity is related to the direct membrane toxicity at distal tubular cells, thus inducing severe distal tubular acidosis and afferent arteriole vasoconstriction with consequent glomerular injury [71]. Liposomal formulations have contributed to minimize the direct cytotoxic effects on epithelial cells and are now routinely employed in the ICU setting [72]. Finally, the use of colistin is dramatically increased in the last years as a lifesaving treatment in sepsis related to multidrug-resistant (MDR) bacteria, despite its known nephrotoxicity. The incidence of colistin-induced AKI varies across studies ranging from 10% up to 55%, as reported in a recent systematic review and meta-analysis [73]. The meta-analysis of 20 studies including 2400 patients comparing the risk of AKI in colistin-based therapy to AKI risk with other antibiotics in the treatment of MDR Gram-negative bacteria demonstrated a significant 82% higher incidence of AKI, although the majority of AKI cases were usually mild and reversible, with a significant proportion not requiring renal replacement therapy. In addition, the combination therapy with a carbapenem could further reduce the impact of colistin-induced AKI [73]. Although renal damage induced by colistin was historically considered as related to the effects of this drug on the cell membrane of proximal tubule cells, latest studies have proposed the importance of other mechanisms, such as the intracellular accumulation and mitochondrial dysfunction [74].

## 4. Pathophysiology of AKI in Kidney Transplantation

KTRs are exposed to additional potential conditions associated with a high risk for AKI, such as ischemia–reperfusion injury, surgical complications, acute antibody mediated or T-cell mediated rejection, side effects from immunosuppressive therapy, urinary tract infections (UTIs) and sepsis [12,75,76]. However, in the context of the transplant setting, chronic immunosuppressive treatment plays a role in impairing the immune response as compared to the general population. Regardless of the etiology of AKI in kidney transplant recipients, there are several pathophysiological mechanisms involved in the disease’s development and course, including hemodynamic and microcirculatory alterations, endothelial dysfunction and tubular cell damage [77]. These processes are characterized, in the first instance, by changes in renal blood flow that may cause reduced graft perfusion and consequent renal ischemia; secondly, ischemic damage may result in a reduced cellular supply of oxygen and metabolic substrates with subsequent cellular damage, production of ROS and activation of inflammation, coagulation and complement pathways [77,78]. Lastly, these mechanisms induced significant impairment in body fluid homeostasis, with reduced GFR and consequent activation of the renin–angiotensin system, resulting in reduced urinary output with subsequent alterations in electrolytes and acid–base balance [79,80]. 

Furthermore, the use of immunosuppressive drugs naturally exposes recipients to a higher risk of contracting infectious diseases: in the context of sepsis, the risk of AKI onset is not only related to hemodynamic alteration and reduced renal blood flow, but due to direct pathogen effects on renal tissue. Several studies have also shown that E. coli infections result in site-specific endothelial dysfunction at the renal proximal tubule through bacterial attachment to the apical tubular cells membrane, with consequent local reduction in peritubular blood flow and increased endothelial permeability [81,82,83]. Specific pathogen-associated molecular patterns (PAMPs) are able to bind Pattern Recognition Receptors (PRRs) located on both immune cells and proximal tubular cells [84,85,86], such as the Toll-like receptors (TLRs), leading to Nuclear Factor kappa B (NF-κB) pathway activation and the production of pro-inflammatory cytokines (such as interleukin-1, interleukin-6 and interferon β) that increase inflammation-related damage, chemokines (such as MCP-1) and adhesion molecules [85,86]. In this context, the release of pro-inflammatory cytokines result in endothelial cell dysfunction, inducing coagulation cascade activation and leukocyte recruitment [87]. In addition, damaged epithelial cells release damage-associated molecular patterns (DAMPs), such as structural and nonhistone high-mobility group box 1 (HMGB1) structural protein and heat shock proteins (HSPs), which may amplify this immunological response and extend graft dysfunction (Figure 2). On the other hand, the described interactions and the production of specific cytokines also stimulate the innate immune response and, through the activation of interstitial dendritic cells, also a subsequent involvement of the adaptive immune response [85]. A further link between innate and acquired immunity is the activation of the complement system [15,88,89]; we recently showed in a porcine model of LPS-induced AKI that there is a marked increase in the deposits of PTX3 protein (which stimulates complement activation by the classical pathway) and C5b-9 (terminal complex produced as a result of activation of the complement cascade) at the peritubular and glomerular capillaries 24 h after LPS infusion [90]. Notably, C5b-9 appears to exert a profibrotic role within renal tissue in this context, as demonstrated by in vitro experiments in which an increase in collagen production by C5b-9-treated human glomerular and tubular epithelial cells was observed, demonstrating the role of complement activation in cellular damage, tissue fibrosis, and inflammation [91,92].

### COVID-19-Associated AKI in Kidney Transplantation

With the advent of the COVID-19 pandemic, KTRs have been exposed to a new risk factor for the development of AKI, as they are more susceptible to severe forms of infection with worse clinical outcomes due to the limited humoral and cellular response to vaccine in this population [93]. In this scenario, KTRs have a high risk of severe complications from COVID-19; thus, the risk of developing AKI is higher in KTRs hospitalized with COVID-19 [57]. Although several risk factors are shared with the general population (older age, obesity, chronic comorbidities), immunosuppressive treatment plays a pivotal role in increasing the risk of worse outcomes in this population. Histological injury during COVID-19 infection was described as extremely heterogeneous, ranging from mild tubular-interstitial injury to histological pictures of collapsing glomerulopathy, thrombotic microangiopathy, focal segmental glomerulosclerosis and, less frequently, other forms of glomerular lesions [94].

SARS-CoV-2 can promote the development of AKI through multiple pathophysiological mechanisms. An indirect mechanism has been firstly reported as the consequence of severe hemodynamic alterations in the context of COVID-19 pneumonia with consequent cardiac dysfunction, decreased cardiac output and hypoperfusion [95]. Common mechanisms include the release of DAMPs and PAMPs at the circulatory level with subsequent local inflammation of renal tissue and activation of the immune response, acute tubular damage, complement activation (with increased C5b-9 deposits at the renal tissue level), promotion of coagulation and endothelial damage [94,96].

The overproduction of proinflammatory cytokines (IL-6, TNF-a, IL-1) induced endothelial dysfunction and other inflammatory pathways similar to those involved in sepsis and leading to multi-organ dysfunction [94]. Another key player in this disease is the immunoregulatory activity of type I interferon; several studies reported that SARS-CoV-2 infection can lead to the suppression of interferon release and that treatment with interferon increased viral clearance, reducing inflammatory markers [97].

In addition, a mechanism of the direct infection of renal cells by SARS-CoV-2 has been hypothesized, as the presence of SARS-CoV-2 was observed in various renal districts, but especially at the glomerular level [98]. Interestingly, the presence of viral RNA within the renal tissue has been demonstrated to be increased in COVID-19 patients with AKI [53]. Some studies have suggested that SARS-CoV-2 is a cytopathic virus able to enter the host cells through the membrane protein Angiotensin I Converting Enzyme 2 (ACE2). The high expression of ACE2 and the Transmembrane Protease Serine 2 (TMPRSS2), colocalized in proximal tubular epithelial cells, therefore represent a potential target of direct kidney injury [94]. In addition, the viral particles can enter and damage podocytes through CD147, inducing cytoskeletal alterations and consequent glomerular damage with proteinuria (diffuse foot processes fusion and histological features of focal segmental glomerulosclerosis) [99]. Moreover, vascular endothelial activation and dysfunction induces a vicious cycle enhancing the release of proinflammatory cytokines and inducing an imbalance between anti- and procoagulant factors, leading to hypercoagulability and increased risk of thrombotic episodes [95]. The microvascular and endothelial damage induced by COVID-19-associated AKI appears to be more pronounced, and platelet activation and thrombin formation also appear to be critical in patients with severe complications [100,101,102]. 

Finally, it is noteworthy to report the other causes of AKI in kidney transplantation should be taken into account; the need to revise the immunosuppressive regimen, such as by modifying and/or stopping immunosuppressive drugs, may lead to changes in the immunological status with donor-specific antibodies (DSA) formation and, consequently, the development of humoral rejection [18].

## 5. Biomarkers of AKI Applied in Kidney Transplantation

Classical AKI criteria are known to present several limitations in assessing kidney disease in the acute setting. Serum creatinine is still the gold standard for defining kidney dysfunction, but it represents a late marker, not able to early identify kidney injury. In addition, several factors may affect creatinine values, particularly in the acute setting of ICU patients; the creatinine production rate varies according to age, gender, muscle mass, diet, and nutritional state. Moreover, a prolonged ICU stay significantly affects muscle mass and, consequently, creatinine values.

Over the years, many biomarkers related to AKI have been studied, aiming not only to diagnose AKI at an early stage but also to identify potential therapeutic targets for treatment [103,104]. NGAL (neutrophil gelatinase-associated lipocalin), KIM-1 (kidney injury molecule 1), TIMP-2 (tissue inhibitor of metalloproteinase 2), IGFBP7 (insulin-like growth factor-binding protein 7) and cystatin C are some of the most studied AKI biomarkers [75]. Among these, TIMP-2 and IGFBP7 are the only FDA-approved AKI biomarkers, targeting G1 cell cycle arrest of tubular epithelial cells and their consequent release at the urinary level following noxious stimuli in several clinical conditions [105,106,107]. Limited data are available to date about the role of these molecules in the context of the kidney transplant patients. Bank et al. performed serial measurements (from day 1 to day 7, day 10, week 6 and month 6 after kidney transplantation) of TIMP-2 and IGFBP7 in 74 KTRs, investigating the performance of these biomarkers to predict the occurrence and duration of DGF after donation after circulatory death (DCD) recipients: TIMP-2 showed better performance in identifying patients with DGF (area under the curve (AUC) 0.91) compared to IGFBP7 [108]. NGAL is a protein that is secreted in the urine and acts as a biomarker of distal tubular damage; this molecule has also been studied in the context of DGF, although its specificity is low, as high urinary levels are also found in patients with UTIs and sepsis [109]. KIM-1 is a type 1 transmembrane protein, which is rapidly expressed in proximal tubules after kidney injury. Few data are available in the setting of kidney transplantation; Filed et al. analyzed urinary KIM-1 in 182 donors, showing higher levels in donors whose kidneys presented DGF after transplantation [110], suggesting the potential application of such biomarkers after kidney transplantation. Moreover, liver fatty-acid binding proteins (L-FABPs) have been widely investigated in the context of preclinical models of AKI and in specific fields (cardiac surgery, sepsis): in the setting of kidney transplantation, Yang et al. investigated urinary L-FABP and NGAL after kidney transplantation and reported the association between 0 h L-FABP values and poor long-term outcomes [111].

More recently, Castellano and colleagues analyzed the role of anti-senescence protein Klotho in the kidney transplant setting: Klotho levels were significantly reduced in DGF patients, suggesting that Klotho deficiency may play a significant role in DGF-associated chronic allograft dysfunction due to its pleiotropic functions (prevention of fibrosis and endothelial dysfunction) [15]. Another noteworthy protein biomarker is Corin, a serine protease implicated in the production of Atrial Natriuretic Peptide (ANP) and playing a protective role against renal damage; plasma levels of Corin are reduced in KTRs with DGF and may serve as a marker of renal ischemia–reperfusion injury [112].

Overall, all of the listed biomarkers have rationales for their application in clinical practice; however, most of them are not well characterized and are based on studies with limited simple sizes. Considering the complexity of the pathophysiology of AKI and kidney transplant setting and the high cost of production and validation, further investigations are required before their routinary application in clinical practice.

### miRNA in AKI and Kidney Transplantation

Transcriptomic biomarkers have also been investigated over the years, among which microRNAs (miRNAs) have been studied in kidney disease and organ transplantation. miRNA dysregulation related to genetic events or inhibition of regulatory enzymes have been reported to be involved in the pathogenesis of several disease, including kidney disease [113]. Their function is not only necessary intracellularly, but they are usually secreted in extracellular vesicles, acting as hormones and mediating paracrine effects [113]. In addition, miRNAs regulate gene expression at the posttranscriptional level, and their stability in biofluids (urine and plasma) make such molecules interesting biomarkers [114].

Several miRNAs play a crucial role in the patient with AKI. For example, increased urinary levels of miR-494, which inhibits the expression of the activating transcription factor 3 (ATF3) gene involved in the NF-κB pathway, have been shown in patients with AKI before the rise in serum creatinine [115]. In addition, increased levels of miR-107 have been described in septic patients with AKI, leading to an increased TNF-α secretion by endothelial cells and consequent tubular damage [116]. Moreover, the inhibition of miR-21 has an anti-apoptotic and inflammation-regulating function in renal tissue. Indeed, it has been observed that inhibition of this miRNA correlates with apoptosis of tubular cells and thus with increased severity of AKI [117]; in addition, miR-21 is associated with increased tissue fibrosis and progression to chronic kidney disease [118]. Ge et al. investigated the different expression profiles of miRNAs in septic-associated AKI as compared to septic patients without AKI and controls, identifying 37 miRNAs differentially expressed; among them, miR-4321 and miR-4270 were significantly upregulated in sepsis-induced AKI and the pathways involved are mainly related to the regulation of oxidative stress and mitochondrial dysfunction [119].

Several miRNAs have been evaluated in the setting of kidney transplantation, with a specific focus on ischemia–reperfusion injury. The upregulation of miR-146a has been described in urine samples of kidney transplant recipients from deceased donors compared to those from living donors, suggesting its role as diagnostic marker for ischemia–reperfusion injury [120]. MicroRNAs are involved in the angiogenetic and apoptotic processes through the effects on different pathways (TGF-b, endothelin, vascular endothelial growth factor, PDGF signaling), creating a chance to discriminate between different graft injuries (acute rejection vs. delayed graft function) [121]. Some of these miRNAs have been reported as being in association with cellular and humoral rejection. Soltaninejad et al. described the upregulation of specific senescence-associated miR-223 and miR-142-3p in the grafts and PBMCs of patients with acute T cell-mediated rejection [122]; similarly, the upregulation of miR-142-5p was described in PBMCs from patients with chronic antibody-mediated rejection [123]. Finally, other miRNAs were investigated in the context of chronic graft dysfunction; lower expressions of miR-200b, miR-211 and miR-204 were described in the urine of patients with interstitial fibrosis and tubular atrophy [124,125].

Overall, although several molecules have been proposed as diagnostic and/or predictive biomarkers for AKI and kidney transplantation, their application in the clinical practice is still lacking due to limited data on the discriminatory performance of such markers. A better characterization of this condition may lead to the identification of new molecules that can not only uniquely identify acute kidney damage and better discriminate different causes of AKI in the transplant setting but can also serve as potential therapeutic targets.

## 6. Graft and Patients’ Outcomes

Long-term graft and patient outcomes are a matter of major concern among KTRs admitted to the ICU. AKI has a higher prevalence in KTRs, and it is associated with an increased risk of graft loss and death by all causes [3]. While ICU admission mainly occurs in the late post-transplant period (probably due to effective preventive strategies in managing KTRs in the early period), studies reported a higher incidence of ICU admission after increased immunosuppression following acute rejection episodes [9,43]. As the risk of AKI in KTRs is higher during ICU stay due to the reasons explained before, clinical presentations and histologic changes may vary, and a lower eGFR at baseline is the best predictor of AKI occurrence [3]. Long-term outcomes on graft and patient survival have been poorly investigated in this setting, although the risk of progression to CKD with consequent increased cardiovascular risk and mortality is suspected to be similar to the non-transplant population. Around 40% of KTRs require renal replacement therapy (RRT) in comparison to one-fifth of non-transplant patients with AKI [18,21]. In addition, the percentage of KTRs discharged alive and free from dialysis is reduced compared to unselected critically ill patients [3]; in addition, a reduced GFR after ICU and hospital discharge is reported in around 25% of cases [10,18]. In a retrospective observational cohort study, amongst 200 KTRs admitted to the ICU, AKI to CKD progression has been estimated to occur in almost half of all survivors at 6 months [18]. The degrees of tubulointerstitial fibrosis and irreversible glomerular damage proper due to chronic kidney injury, however, may be mitigated by the mechanisms of polyploidization-mediated cell hypertrophy and the progenitor proliferation of tubular cells and podocytes [126,127]. In addition, the role of complement activation during AKI may promote tubular epithelial cells senescence and the development of a senescence-associated secretory phenotype (SASP) that may be involved in renal aging and fibrosis [128]. Hospital mortality is reported to be higher in this specific population, with described rates up to 30%, particularly in critically ill patients with severe septic shock, mechanical ventilation and multiorgan dysfunction [3]. Regardless of etiology, AKI is associated with higher risk of graft loss (hazard ratio, HR 2.74, 95%CI 2.56–2.92), death with a functional transplant (HR 2.36, 95%CI 2.14–2.60) and death-censored graft loss (HR 3.17, 95%CI 2.91–3.46) [19]. 

KTRs with AKI showed an increased risk of cardiovascular disease and mortality with respect to KTRs without AKI. A recent retrospective, single-center cohort study reported that AKI is associated with a 38% increased risk of major adverse cardiovascular events (MACE) and an 86% increased risk of cardiovascular mortality [129]. However, pre-existing cardiovascular diseases before kidney transplantation may represent a risk factor for ICU admission and mortality rate among this population. In line with this hypothesis, Lenihan et al. showed that history of atrial fibrillation significantly increases the risk of post-transplant death, graft loss and ischemic stroke compared to KTRs without atrial fibrillation. Furthermore, arterial hypertension may ensue following AKI, possibly being an indirect sign of latent CKD. In the non-transplant population, one-fifth of AKI survivors will suffer from arterial hypertension [80]. Although arterial hypertension is a well-described complication of kidney transplantation, data regarding its association with AKI in this subpopulation are sparse and inconsistent; further research is warranted to confirm the importance of this interplay in this setting. Finally, donor-specific antibodies (DSA) play a critical role in graft survival after ICU admission; in fact, the need for modulating immunosuppressive therapy and red blood cell transfusions influence the immunological outcome of KTRs admitted to the ICU. Guinault et al. have shown that among 119 KTRs with available anti-HLA data before and 6 months after ICU admission, 18 patients (15.1%) developed anti-HLA antibodies, with 9 patients (9.2%) developing DSA; 6 of these did not present antibodies before ICU admission [18]. Similarly, Masset et al. investigated the de novo occurrence of DSA in COVID-19-infected KTRs, estimating their incidence at 4%, limited to patients with significantly reduced immunosuppressive therapy and high immunologic risk profiles [130].

Finally, limited data are available to date about the application in the AKI and kidney transplant setting of novel therapeutic approaches (mineralocorticoid receptor antagonists, SGLT2 inhibitors, etc.) characterized by the ability of reducing proteinuria and increased cardio- and renal protection in the general population [131]. Recently, Sánchez-Fructuoso et al. reported the results of a multicentric observational study, including 339 diabetic KTRs treated with SGLT2i, demonstrating several positive effects (decrease in HbA1c; reduction in body weight, blood pressure and serum uric acid; and increase in serum magnesium levels and hemoglobin) even in the kidney transplant setting, without significant side effects (no increased risk for UTIs compared to the general population) [132].

## 7. AKI Management in KTRs

Despite the great efforts made in achieving a better understanding of the pathophysiological mechanisms involved in AKI, the management of such a condition is still limited to general measures, including volume resuscitation, hemodynamic management for optimization of hypovolemic conditions and early initiation of a sepsis care bundle with prompt antibiotic treatment in the case of sepsis and septic shock, taken for unselected critically ill patients [28,133,134]. However, KTRs represent a peculiar population whose frailty derives from a single functioning kidney, immunosuppression and hemodynamic-mediated susceptibility to AKI [33].

The diagnostic work-out depends on the clinical conditions and the suspected underlying cause of AKI, as well as on the duration and severity of renal impairment. A regular monitoring of the kidney function and electrolyte plasma levels is mandatory for a proper management of the transplanted patient admitted to the ICU. As a general rule, any increase in serum creatinine exceeding 20% from baseline deserves an attentive evaluation to rule out any possible causes of acute illness and requires a nephrological consultation [135]. In addition, proteinuria evaluation must be routinely performed, since it implies glomerular and tubular damage, along with pro-inflammatory damage of the tubular lining epithelium due to increased mechanical stress [136]. A point-of-care ultrasound assessment (POCUS) and Doppler evaluation are fundamental to exclude acute prerenal causes, including perfusion defects (i.e., renal artery stenosis) and/or obstructive causes of AKI. A recent study demonstrated the high diagnostic yield of resistivity index and power Doppler grading for ATN in kidney allografts on the third day after kidney transplantation [137]. A detailed review of serum levels of immunosuppressive drugs is critical in addition to the evaluation of the immunological status (search for de novo HLA antibodies and/or DSA), as it may lead to the need of a graft biopsy to rule out acute rejection [138]. Moreover, a complete review of all medications is required, and all nephrotoxic drugs should be suspended when possible and those with renal excretion adjusted according to the actual GFR [138].

RRT initiation may be required in half of KTRs with severe AKI. In this case, when a previous dialysis access is not available, a new vascular access should be placed, taking care to avoid the placement of a femoral catheter on the same side of the graft. Hemodynamic instability and the need for optimal volume status control are the leading factors for the use of continuous renal replacement therapies (CRRT) instead of the intermittent modalities. The choice of the most appropriate CRRT modalities is subordinated to the patient’s characteristics, healthcare providers’ expertise and cost-effectiveness of the treatment; thus, a multidisciplinary approach involving intensivists, nephrologists and transplant teams is strongly recommended [71]. There is no consensus concerning the optimal dose of CRRT to be delivered to ICU-admitted KTRs. Data from the RENAL Replacement Therapy Study randomized controlled trial reported the lack of statistically significant differences between intensive and conventional therapy with CRRT among unselected critically ill patients [139].

### Management of Immunosuppression in KTRs with AKI

The optimal management of immunosuppressive therapy in KTRs admitted to the ICU may be pivotal in assessing clinical outcomes. There are several different immunosuppressive protocols with different combination of drugs, based on the individual risk of rejection as well as potential side effects to a specific drug. The foremost recommended combination of immunosuppressants for maintenance treatment includes a CNI, an anti-metabolite (typically mycophenolate mofetil or the enteric-coated, delayed-release formulation of mycophenolic acid) and low-dose oral prednisone [138]. Nonetheless, a proper modulation of immunosuppressive therapy in KTRs admitted to the ICU may be challenging for intensivists and requires a strong collaboration with nephrologists and transplant team to adequately balance the risk of rejection to the risk of life-threatening conditions (sepsis and septic shock) [12]. In addition, drug interactions are common between immunosuppressive agents and antibiotics and other drugs typically used in ICUs. For example, fluconazole, macrolides, diltiazem and verapamil are typical drugs that increase CNI concentration, while others (rifabutin, rifampicin, phenytoin, barbituric acid) decrease CNI trough levels. It is therefore important to check for these potential interactions and adequately monitor the dose accordingly. In addition, clinical conditions characterized by shock and consequent acute renal impairment may significantly alter drug pharmacokinetics and pharmacodynamics, requiring a close monitor of trough levels.

No consensus is available to date on the optimal management of immunosuppression in such conditions, and specific therapeutic adjustments, taking into consideration the underlying etiology of AKI in the ICU setting, are required. Figure 3 proposes an algorithm for managing immunosuppression during ICU stay. Moderate-to-severe sepsis require a reduction in and/or withdrawal of the antiproliferative immunosuppressant, with concomitant lowering of CNI trough levels. Maintenance of low doses of oral corticosteroids or switching to intravenous corticosteroids is advisable. In KTRs with severe sepsis and septic shock, all immunosuppressants should be stopped with the introduction of intravenous corticosteroids with the aim to avoid any undesired additional immunosuppression in such critical condition. Additional steroids for sepsis treatment are still object of controversies in the ICU bundle of care and are not recommended to date [133]. For candidates for major surgical interventions, it is advisable to temporarily switch mTOR inhibitors to CNI one week prior to surgery (up to two weeks for sirolimus), due to their antifibrotic effect that may hinder a proper wound healing [135]. Similarly, the potential re-introduction of immunosuppressive treatment should be evaluated based on the patient’s condition and comorbidities.

## 8. Conclusions

In conclusion, AKI following ICU admission in KTRs represents a major concern for nephrologists, as those patients are at higher risk of allograft loss and mortality. Sepsis represents the main cause of ICU admission and AKI development with pathophysiological mechanisms similar to non-transplant patients with sepsis; however, the immunological status of KTRs increases the risk of progression to severe forms of diseases and mortality that affect up to 30% of KTRs admitted in ICU. In this scenario, an optimal management should include volume status optimization, hemodynamic monitoring and prompt and appropriate antibiotic treatment; in addition, immunosuppression should be carefully managed according to patients’ condition and required a strong collaboration between nephrologists and intensivists.

## Figures and Tables

**Figure 1 biomedicines-11-01474-f001:**
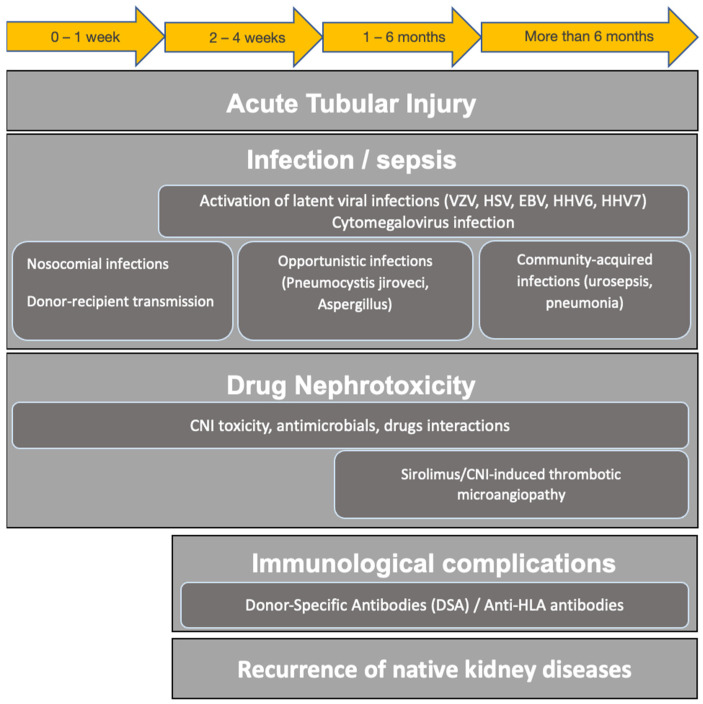
Main causes of AKI in kidney transplantation. **Legend.** CNI, calcineurin inhibitors; EBV Epstein–Barr virus; HLA, human leukocytes antigens; HHV6, human herpesvirus 6; HHV7, human herpesvirus 7; VZV, varicella zoster virus.

**Figure 2 biomedicines-11-01474-f002:**
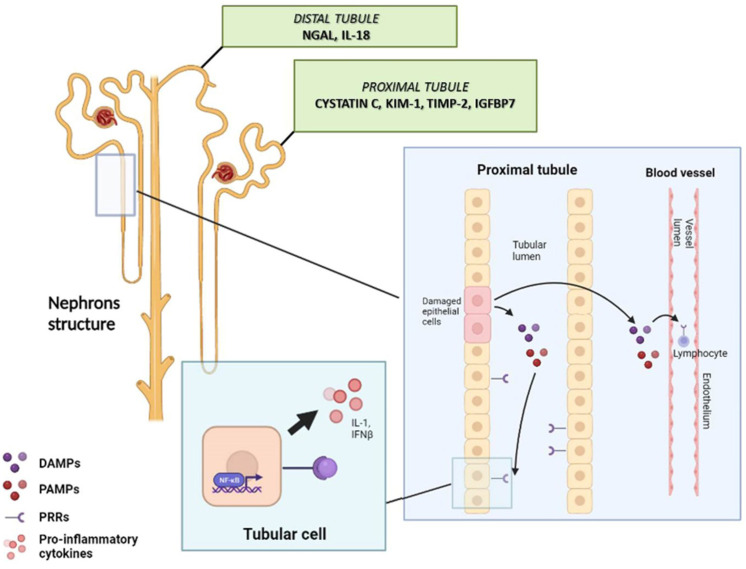
Mechanisms and biomarkers involved in inflammatory response and tubular damage in AKI in kidney transplantation. **Legend**. DAMPs: damage-associated molecular patterns; KIM-1, kidney injury molecule-1; IFNβ, interferon β; IGFBP7, insulin-like growth factor binding protein 7; IL-1, interleukin 1; IL-18, interleukin 18; NGAL, neutrophil gelatinase-associated lipocalin; PAMPs: pathogen-associated molecular patterns; PRRs: pathogen-associated molecular patterns; TIMP-2, tissue inhibitor metalloproteinase 2.

**Figure 3 biomedicines-11-01474-f003:**
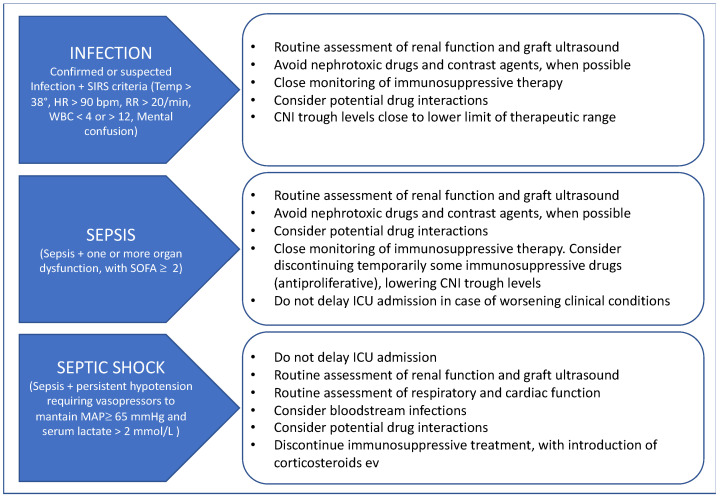
Management of immunosuppression in critically ill KTRs with AKI. Legend: CNI, calcineurin inhibitors; HR, heart rate; ICU, intensive care unit; MAP, mean arterial pressure; RR, respiratory rate; SIRS, Systemic Inflammatory Response Syndrome; SOFA, Sequential Organ Dysfunction Assessment; WBC, white blood count.

## Data Availability

Not applicable.

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
