# Peer review of "Acute Kidney Injury in Kidney Transplant Patients in Intensive Care Unit: From Pathogenesis to Clinical Management"

_biomedicines, 2023, doi:10.3390/biomedicines11051474_

Round 1

Reviewer 1 Report

A very interesting and well-described review; just one recommendation, considering the recent COVID-19 pandemic, it would be nice if you could add a section dedicated to SARS-CoV-2 infection in renal transplant patients (throughout your article, you presented different data related to this topic, but in my opinion, it should be better emphasised in a separate section).

Author Response

We thank the reviewer for their very careful reading of our manuscript and for his/her constructive comments. Reviewer’s concerns have been addressed point-by-point as follows.

A very interesting and well-described review; just one recommendation, considering the recent COVID-19 pandemic, it would be nice if you could add a section dedicated to SARS-CoV-2 infection in renal transplant patients (throughout your article, you presented different data related to this topic, but in my opinion, it should be better emphasised in a separate section).

  • We thank the reviewer for this suggestion, that is in line with the suggestion of the reviewer 2. Covid-19 pandemic represented a major health problem worldwide and we agree that it deserves to be mentioned in a dedicated section.

Reviewer 2 Report

I measured the manuscript entitled “Acute Kidney Injury in Kidney Transplant Patients in Intensive Care Unit: from pathogenesis to clinical management” by Marco Fiorentino, et al, that is intended to be published in Biomedicines journal.

I enjoyed reading the manuscript, the authors have made a considerable effort to summarize everything related to the subject. However, few new data is given in this old subspeciality of nephrology. Majority of the concepts were steadily described in the eighties and nineties. It appears as a chapter of a book of Nephrology, or if we are more considered a chapter of a book of renal transplantation.

To me, here, the Intensive Care Unit environment is just circumstantial. All the epidemiologic, diagnostic challenges and treatment management occurs more frequently in the regular hospitalization area.

And all the specific mechanisms related to renal transplantation, as it is cell alloresponse or HLA antibodies are only marginally commented. Description does not appear as problems around renal transplantation but around acute kidney injury in general.

To me there are only some innovative new concepts:

- Current data concerning Covid19

- Their contribution in the mechanisms of acute renal failure: PMMA-Based Continuous Hemofiltration Modulated Complement Activation and Renal Dysfunction in LPS-Induced Acute Kidney Injury. Front Immunol 2021, 12, 605212, doi:10.3389/fimmu.2021.605212.

-  The discussion of miRNA as new Biomarkers. A comprehensive and more extensive manuscript related to this subject appears as an interesting novel review.

Author Response

We thank the reviewer for their very careful reading of our manuscript and for his/her constructive comments. Reviewer’s concerns have been addressed point-by-point as follows.

I measured the manuscript entitled “Acute Kidney Injury in Kidney Transplant Patients in Intensive Care Unit: from pathogenesis to clinical management” by Marco Fiorentino, et al, that is intended to be published in Biomedicines journal. I enjoyed reading the manuscript, the authors have made a considerable effort to summarize everything related to the subject. However, few new data is given in this old subspeciality of nephrology. Majority of the concepts were steadily described in the eighties and nineties. It appears as a chapter of a book of Nephrology, or if we are more considered a chapter of a book of renal transplantation. To me, here, the Intensive Care Unit environment is just circumstantial. All the epidemiologic, diagnostic challenges and treatment management occurs more frequently in the regular hospitalization area. And all the specific mechanisms related to renal transplantation, as it is cell alloresponse or HLA antibodies are only marginally commented. Description does not appear as problems around renal transplantation but around acute kidney injury in general.

To me there are only some innovative new concepts:

- Current data concerning Covid19

- Their contribution in the mechanisms of acute renal failure: PMMA-Based Continuous Hemofiltration Modulated Complement Activation and Renal Dysfunction in LPS-Induced Acute Kidney Injury. Front Immunol 2021, 12, 605212, doi:10.3389/fimmu.2021.605212.

-  The discussion of miRNA as new Biomarkers. A comprehensive and more extensive manuscript related to this subject appears as an interesting novel review.

  • We thank the reviewer for the comments. In our opinion, although several aspects are in common between AKI in the transplant and not-transplant settings, kidney transplant recipients are also characterized by specific features that need to be considered in the clinical management of these patients; in addition, epidemiological data of such condition as well as the different causes of AKI in this setting should be well known for an appropriate management of these patients, particularly in the context of hospitals without nephrologists or a kidney transplant center.
  • We thank the reviewer for pointing out some innovative concepts introduced with this review. As suggested, we added a more detailed section dedicated to miRNA as new biomarkers of AKI in kidney transplant recipients.

Reviewer 3 Report

The review is interesting and well-written.

I have the following comments:

1.      A comment is required in the text regarding Figure 1. The depicted in many textbooks time frame needs some clarification. For instance, although CMV infection typically happens early after transplantation, nowadays, most protocols include preventing anti-CMV therapy. As a consequence, many CMV infections occur later. As another example, typically, most kidney diseases recur relatively late, but some of them, such as the FSGN, can happen in the first post-transplantation days. 

2.      In the subsection of drug nephrotoxicity, a notion about the antibiotic colistin, which is increasingly used for gram-negative infections, especially in the ICU, would be welcome.

3.      In the subsection on biomarkers, a notion about the classical marker of serum creatinine would be helpful. Since many kidney transplant recipients receive corticosteroids for a long time, they might have reduced muscle mass, making their risk stratification (previous creatinine-based eGFR) for AKI problematic. 

Author Response

We thank the reviewer for their very careful reading of our manuscript and for his/her constructive comments. Reviewer’s concerns have been addressed point-by-point as follows.

The review is interesting and well-written.

I have the following comments:

  1. A comment is required in the text regarding Figure 1. The depicted in many textbooks time frame needs some clarification. For instance, although CMV infection typically happens early after transplantation, nowadays, most protocols include preventing anti-CMV therapy. As a consequence, many CMV infections occur later. As another example, typically, most kidney diseases recur relatively late, but some of them, such as the FSGN, can happen in the first post-transplantation days.

  • We thank the reviewer for this comment. We agree with the suggestions raised by the reviewer; we understand that CMV reactivation may occur later than 1 year after kidney transplantation and this is the reason why the related box included the time window “more than 6 months”, just to consider the potential impact of such diseases during long-term follow up. We agree with the potential recurrence of specific native kidney disease (FSGS, thrombotic microangiopathy); accordingly, we modified the figure 1, increasing the length of the box from the time window “2-4 weeks” to “more than 6 months”.

  1. In the subsection of drug nephrotoxicity, a notion about the antibiotic colistin, which is increasingly used for gram-negative infections, especially in the ICU, would be welcome.

  • We agree and thank the reviewer for the suggestion, and we added some comments on colistin treatments.

  1. In the subsection on biomarkers, a notion about the classical marker of serum creatinine would be helpful. Since many kidney transplant recipients receive corticosteroids for a long time, they might have reduced muscle mass, making their risk stratification (previous creatinine-based eGFR) for AKI problematic.

  • We agree with the reviewer and revise the section, including some comments to the classical markers of kidney function and their potential limitations and expanded the section dedicated to miRNA as innovative biomarkers in this setting, as suggested by reviewer 2.

Reviewer 4 Report

This is a nice paper. However, I have some comments. The findings from this paper are excellent and worthy to review. This manuscript contained some questions described below. I think this paper is interesting, this review contributes to future's clinical medicine largely. I have some questions from a point of view of clinical medicine. What are the biomarkers of AKI? Urinary L-FABP is an early biomarker for tubular ischemia, but I cannot find an explanation for this biomarker. Please add this information as it seems to be an important biomarker. Also what are the potential drugs that could prevent AKI? Please tell us. For example, please mention ARNI, SGLT2 inhibitors, HIF-PH inhibitors.

Author Response

We thank the reviewer for their very careful reading of our manuscript and for his/her constructive comments. Reviewer’s concerns have been addressed point-by-point as follows.

This is a nice paper. However, I have some comments. The findings from this paper are excellent and worthy to review. This manuscript contained some questions described below. I think this paper is interesting, this review contributes to future's clinical medicine largely. I have some questions from a point of view of clinical medicine. What are the biomarkers of AKI? Urinary L-FABP is an early biomarker for tubular ischemia, but I cannot find an explanation for this biomarker. Please add this information as it seems to be an important biomarker. Also what are the potential drugs that could prevent AKI? Please tell us. For example, please mention ARNI, SGLT2 inhibitors, HIF-PH inhibitors.

  • We thank the reviewer for this suggestion. We expanded the section related to AKI biomarkers, mainly focusing on those applied on the kidney transplant setting. About AKI prevention, to date AKI prevention is mainly based on supportive care, based on volume control, hemodynamic stability and avoiding nephrotoxicity. The advent of new treatment options affects more the chronic setting than the acute one, so SGLT2 inhibitors demonstrated a significant reduction in the risk of cardiovascular diseases and progression to ESRD in patients with chronic kidney disease independently of the presence of diabetic kidney disease (DKD). In the kidney transplant setting, available data are limited to assess their benefit in this cohort of patients. We added a comment on this point in the section on graft outcomes.

Round 2

Reviewer 2 Report

I saw two other reviewers more or less accepted the MS, and one third asked for major revision. I still believe the manuscript is just a plagiarism (not word by word) of old reviews or a chapter of a book.

Reviewer 3 Report

The authors addressed all issues.